# Field Test of the Propheromones of the Whitemarked Tussock Moth (WMTM) *Orgyia leucostigma* (J.E. Smith) (Lepidoptera: Erebidae)

**DOI:** 10.3390/insects14110880

**Published:** 2023-11-15

**Authors:** Peter Mayo, Sumudu Deepa Abeysekera, Peter J. Silk, David I. MaGee, Gaetan Leclair, Jon Sweeney, Jeffrey Ogden

**Affiliations:** 1Atlantic Forestry Centre, Canadian Forest Service, Natural Resources Canada, Fredericton, NB E3C 2G6, Canadajon.sweeney@nrcan-rncan.gc.ca (J.S.); 2Chemistry Department, University of New Brunswick, Fredericton, NB E3B 5A3, Canada; dmagee@unb.ca; 3Nova Scotia Department of Natural Resources and Renewables (NSDNRR), Halifax, NS B3J 2T9, Canada; jeffrey.ogden@novascotia.ca

**Keywords:** whitemarked tussock moth, propheromones, improved detection device, (*Z*,*Z*)-6, 9-heneicosadien-11-one, (*Z*,*Z*)-11, 11-dimethoxy-6, 9-heneicosadiene, (*Z*,*Z*)-6,9-heneicosadien-11-one ethylene ketal

## Abstract

**Simple Summary:**

The whitemarked tussock moth is a pest of balsam fir in the Maritime provinces of Canada, and also poses a threat to human health because the hairs on the larvae are urticating (i.e., they irritate the skin). In this manuscript, we improve a known lure, based on the pheromone of this insect, with the aim to use it as a widespread monitoring/detection method. Because the female-produced sex pheromone of this moth is unstable, lures containing this pheromone only last a day or two in the field; however, our detection method takes advantage of a synthetic propheromone, or pheromone precursor, that is converted to the pheromone by acid within the lure and releases it over time, circumventing the problem of its instability. We found that our lure formulation and pheromone delivery method is more effective as a lure for whitemarked tussock moth than the control treatments in our field study.

**Abstract:**

The whitemarked tussock moth (WMTM), *Orgyia leucostigma* (J.E. Smith) (Lepidoptera: Erebidae), is an economic pest in Nova Scotia, Canada, where it undergoes periodic outbreaks defoliating several tree species of economic value, including balsam fir, *Abies balsamea* (L.) Miller (Pinaceae). Herein is described a releasing device for the WMTM sex pheromone (*Z*,*Z*)-6, 9-heneicosadien-11-one based on a rubber septum, which converts pheromone precursors, such as acetals, namely (*Z*,*Z*)-11,11-dimethoxy-6,9-heneicosadiene and (*Z*,*Z*)-6,9-heneicosadien-11-one ethylene ketal, to the pheromone itself by the action of acetic acid and releases it over time. The pheromone is unstable in nature and, consequently, lures made with this compound will only attract WMTM for a day or two. The two pheromone precursors, however, are more stable, and are converted slowly into the pheromone by acetic acid impregnated in the releasing device, or by acidic conditions in the environment. The two pheromone precursors were synthesized in 2019 using a modified, previously published approach. Field trapping studies conducted from 2019–22 showed that traps baited with rubber septa loaded with either (*Z*,*Z*)-11,11-dimethoxy-6,9-heneicosadiene by itself or (*Z*,*Z*)-6,9-heneicosadien-11-one ethylene ketal plus acetic acid consistently caught significantly more WMTM than traps baited with blank septa in most experiments.

## 1. Introduction

The whitemarked tussock moth (WMTM), *Orgyia leucostigma* (J.E. Smith) (Lepidoptera: Erebidae), is a known defoliator of both hardwood and softwood forests of the Acadian and boreal forests of Atlantic Canada. Historically, WMTM has been an economic pest within the provinces of Nova Scotia and New Brunswick, causing severe damage to balsam fir, *Abies balsamea* (L.) Miller (Pinaceae) [1,2]. Previously, field monitoring of this insect has involved large-capacity Multipher traps and a commercial lure loaded with (*Z*)-6-heneicosen-11-one (**1**) (Figure 1), a sex pheromone component of WMTM [2,3]. This compound, however, is only weakly attractive to WMTM males by itself. The diunsaturated compound (*Z*,*Z*)-6,9-heneicosadien-11-one (**2**) (Figure 1) is a much more effective lure for WMTM, and part of the natural pheromone blend [1]. However, it is unstable under natural conditions, and baited traps become unattractive within 1–2 days [2,4]. Compound **2** is also a major sex pheromone component of the painted apple moth *Teia anartoides* [5,6] and one of the three pheromone components of the tea black tussock moth, *Dasychira baibarana* (Lepidoptera: Erebidae: Lymantriinae) [7]. In fact, Grant [8] showed that an extract of female WMTM sex pheromone glands was highly attractive to males but lost its potency after 1–3 weeks. To overcome the instability problem of the pheromone, Grant et al. [2,4] reported a releasing device which converted a synthetic pheromone precursor, or propheromone, to (*Z*,*Z*)-6,9-heneicosadien-11-one (**2**) (the active molecule) in situ and released it over time. This releasing device was based on a miniature autonomous pump that delivered the precursor to an absorbent substrate wetted with a hydrolyzing solution housed in a ventilated vial.

Mayo et al. [9] reported an improved synthesis of the pheromone precursor **3b**, [(*Z*,*Z*)-6, 9-heneicosadien-11-one ethylene ketal, Figure 1, hereafter precursor B], along with an additional pheromone precursor (*Z*,*Z*)-11,11-dimethoxy-6,9-heneicosadiene (Figure 1, **3a**, hereafter precursor A).

Consequently, in the present work, new release devices were designed to allow conversion of the precursors A and B in situ to the dienone **2,** and were field-tested for their efficacy in detecting WMTM in trapping bioassays in four consecutive years. The objectives of this project were twofold: to develop release devices that can be loaded with a pheromone precursor and an acid catalyst and be easily deployed in the field as trap lures, and to determine their efficacy for detecting WMTM in field bioassays, when loaded with pheromone precursors A or B. The detection method proposed here is simpler and easier to deploy in the field than the releasing device proposed by Grant et al. [2,4].

## 2. Materials and Methods

### 2.1. Release Devices: Septum-Based Releasing Device

The syntheses of the two pheromone precursors A and B have previously been reported [9]. Two different kinds of release devices were designed: one is based on red rubber septa (prewashed, Wheaton, DWK Life Sciences, Millville, NJ, USA), and the other is based on coupled vials separated by a semi-permeable membrane. The red rubber septa were soaked in a 5% solution of glacial acetic acid in dichloromethane for 1 h, then air-dried for 3 h. The septa that were used for the acetic acid control were not modified further after this. For lures A and B, 2.00 mg/lure of pheromone precursor A or B, respectively, were pipetted as 200 µL of a 10 mg/mL solution in dichloromethane onto acetic acid-treated septa. To prepare acetic-acid free pheromone precursor controls A and B, 2.00 mg/lure of A or B, respectively, were pipetted as 200 µL of a 10 mg/mL solution in dichloromethane onto clean, untreated septa. The pheromone precursor solutions were allowed to completely soak into the septa (approximately 1 h), air-dried for 3 h, and placed in static-shielding Ziplock^®^ bags (Uline Canada, Milton, ON, Canada). Blank septa controls were simply the clean septa after 200 µL of dichloromethane was pipetted onto them and allowed to soak for 1 h before air-drying for 3 h. All lures were shipped to Nova Scotia, Canada on ice for field testing in 2019–2022.

### 2.2. Vial-Based Releasing Device

This consisted of two polymer screw cap vials (Agilent part 5190-2242) A and C, coupled by two plastic screw caps (the Agilent part welded together head-to-head (B) (Figure 2)). A 0.16 mm thick disk of polyethylene sheet was placed between the two during the weld to make up a permeable membrane. Vial A had four 1/8-inch-wide breather holes drilled upwards into and through the insert part of the vial to allow diffusion of the active component. Vial A is not meant for liquids. Vial C, meant to contain the acid activator, acetic acid in this case, had a hanger ring welded to the underside as it would become the top once assembled. Vial C would be sealed with a screw cap and septa (Agilent part 5182-0717) until deployment time. We tested three vial-based lures in 2021 only: (1) 50 µL of acetic acid in vial C, no pheromone precursor A in vial A; (2) 2 mg of pheromone precursor A in vial A and no acetic acid in vial C; and (3) 2 mg of pheromone precursor A in vial A and 50 µL of acetic acid in vial C.

### 2.3. Trapping experiments

We conducted five different experiments in Nova Scotia during August–September from 2019 to 2022, testing the effects of various lure treatments on trap catches of WMTM (Table 1). Multipher II traps (Solida, Saint-Ferréol-les-Neiges, QC, Canada) were used each year, supplemented with Multipher I traps in 2021. See the following link for illustrations of Multipher I and II traps: https://solida.quebec/produit/multipher-i-ii-and-iii-product-no-40bc001/?lang=en (accessed on 1 October 2023). Traps were spaced 20 m apart in four linear transects of 6–9 traps (depending on the number of lure treatments) per field site. Treatments were assigned randomly among traps within each transect. Traps were checked weekly to record the numbers of captured WMTM, replace old lures with fresh ones, and re-randomize the lure treatments among trap positions. Treatments were replicated both spatially and temporally in completely randomized blocks with each linear transect-trapping week considered a replicate.

### 2.4. Study Area

Trapping experiments were conducted in mixed coniferous stands at nine sites in Nova Scotia during August–September from 2019 to 2022 (Table 1). Sites were selected using a combination of aerial surveillance and site visits. Due to fluctuating WMTM population levels, it was necessary to change the study areas during the experiment to ensure adequate moth totals.

Study sites in 2019 and 2020 included Five Islands Provincial Park, Colchester County (45.40028 N 64.03539 W), Stillwater, Guysborough County (45.19097 N 61.98258 W) and Earltown, Colchester County (45.58323 N 63.11127 W). In 2021, sites were located at Proudfoot Road, Kemptown, Colchester County (45.50028 N 63.02873 W), Dalhousie Mountain, Pictou County (45.56737 N 63.01150 W) and Biorachan Road, Earltown, Colchester County (45.57520 N 63.11741 W). In 2022, two sites were considered in Greenfield, Colchester County; 595 Old Greenfield Road (45.37466 N 63.16633 W) and 1175 Old Greenfield Road (45.37256 N 63.14253 W), and a third site off the Riversdale Road, Riversdale, Colchester County (45.431049 N 63.059932 W).

**Experiment 1** tested the following six lure treatments: (1) Lure A = pheromone precursor A + acetic acid; (2) Lure B = pheromone precursor B + acetic acid; (3) pheromone precursor A; (4) pheromone precursor B; (5) acetic acid; and (6) blank septa. This experiment was conducted at seven different field sites in Nova Scotia: three sites in 2019, one in 2020, one site in 2021, and three sites in 2022 (Table 1). In 2019, 2020 and 2022, Multipher II traps were used, and in 2021, a mix of Multipher I and II were used.

**Experiments 2A and 2B** assessed the lifespan of the lure formulations. Lures were prepared, aged 1 week (Experiment 2A, 2020) or 5 days (Experiment 2B, 2021), then deployed to gauge their effectiveness compared to fresh lures. Each experiment was replicated at one site in 2020 or 2021 (Table 1).

**Experiment 3** was identical to experiment 1, but with the addition of a seventh treatment, the commercially available WMTM lure (Great lakes IPM, Vestaburg, MI, USA). Experiment 3 was replicated at one site in 2020 (Table 1).

**Experiment 4** included the six lure treatments of experiment 1 plus three vial-based lures, and was replicated at one site in 2021 (Table 1).

### 2.5. Statistical Analysis

We used generalized linear models (SAS PROC GLIMMIX) to test the effects of lure treatment, site–year, and block (nested within site–year) on the mean weekly catch of WMTM per trap. Any blocks (i.e., trap transect–week) with zero catches in all lure treatments were omitted from analysis. For experiment 1, we first ran a preliminary analysis to test for significant interactions between lure treatment and site–year, using the model: y = lure + site–year + lure*site–year + block(site–year), with both lure and site–year considered as fixed effects and blocks as random effects. When the site–year interaction was not significant, data were pooled and analyzed using the following model: y = lure + site–year + block(site–year), with lure treatments fixed and both site years and blocks random. Data from experiment 1 were pooled and analyzed using the following model: y = lure + site–year + block(site–year), with lure treatments fixed and both site–years and blocks random. For experiments conducted in one site–year only, we used the model: y = lure + block, with lure treatments fixed and blocks random. Generalized linear models using Gaussian (on both raw data and data transformed by log(y + 1)), Poisson, and negative binomial distributions were run, and results are reported from the model-distribution with the lowest value for the corrected Akaike Information Criterion (AICc). Least-square means were compared using the Tukey–Kramer method with experiment-wise error controlled at α = 0.05, but means and standard errors are reported on raw data. When the generalized linear models would not converge (Experiment 2A, 2B) we used the nonparametric Friedman test on count data converted to ranks, followed by the Ryan–Einot–Gabriel–Welsh multiple range test. We also tested whether the proportion of traps that captured at least one WMTM, i.e., “positive” traps (data from all site years pooled), was independent of lure treatment using Cochran’s Q test, and tested for difference between nine specific pairs of lure treatments, e.g., lure A vs. blank septa, using a Sheffé-like multiple comparison test with α = 0.0056 (Bonferroni-corrected *p* value of 0.05) [10].

## 3. Results

Lure treatment significantly affected the mean trap catch of WMTM in experiment 1, but the relative performance of lures varied among years in which the experiment was conducted. There was a significant interaction between lure treatment and site–year when data from all site–years were pooled (*F* = 4.09; df = 23, 205; *p* < 0.0001). We were able to pool data from all sites from 2019 and 2020 (*F* = 1.52; df = 11, 100; *p =* 0.14) and separately pool the data from all sites in 2022 (*F* = 1.15; df = 6, 105; *p* < 0.25), but data from 2021 (one site—Biorachan) had to be analyzed separately.

Traps baited with lure A (precursor A plus acetic acid) performed better than all other lures in 2019 and 2020, whereas precursor A alone performed the best in 2021 and 2022. Those traps baited with either precursor B or lure B (precursor B plus acetic acid) consistently captured significantly more WMTM than traps baited with acetic acid or blank septa controls in all site–years (Table 2). In 2021, traps baited with precursor B captured more WMTM than traps baited with acetic acid or blank septa, while traps baited with lure A did not.

The proportion of traps positive for WMTM was significantly affected by lure treatment in experiment 1 (Cochran’s Q = 119, *p* < 0.0001). Traps baited with lure A or lure B detected WMTM in 70% and 56% of block replicates, respectively, significantly more than acetic acid (3.7%) or blank septa (0%), but not more than their respective precursors (precursor A, 74%; precursor B, 20%) (Sheffé’s test, *p* < 0.05). Traps baited with precursor A detected WMTM significantly more often than traps baited with blank septa, but the same was not true for precursor B.

In experiment 2A, lure B was the only lure treatment that captured significantly more WMTM than blank septa or acetic acid and detected WMTM in 63% of replicates. Traps baited with lure A or lure B that had been aged 7 days at room temperature captured no WMTM (Table 3). In experiment 2B, traps baited with precursor A or precursor A aged 5 days were the only treatments that caught significantly more WMTM than traps baited with blank septa; both lures detected WMTM in 100% of replicates (Table 3).

In experiment 3, traps baited with lure B captured significantly more WMTM than traps baited with any other lure treatment except the commercial WMTM lure; however, the mean catch in the latter was not significantly different from that in traps baited with acetic acid or blank septa (Table 4). The effect of lure treatment on proportion of traps positive for WMTM was not significant (Cochran’s Q = 11.65, *p* > 0.05).

In Experiment 4, traps baited with the vial-based lure loaded with both precursor A by itself (PA) and acetic acid by itself (AA) detected WMTM in 50% of the replicates, whereas traps baited with either precursor A or precursor B (in septum-based lures) detected WMTM in 100% of replicates. Traps baited with precursor A loaded in septa had the greatest mean catch, significantly more than all other lure treatments except precursor B and lure B. The mean catch in traps baited with the vial-based lure loaded with both precursor A and acetic acid (PA + AA) was not significantly different from that in traps baited with lure B and precursor B, but also no different from that in traps baited with blank septa (Table 5).

Please see supplementary materials for all of the data for the field trapping studies from 2019–2022 (Appendix A).

## 4. Discussion

Propheromones, or molecules that are converted from a precursor to the pheromone in situ in the lure, appear to be a potential method of improving the release of pure pheromone components in different field applications [11]. They can be used when the pheromone itself is unstable under field conditions, or difficult to handle. Another example of the use of an acetal-based sex propheromone can be found in the work of Grodner et al. [12] on an attractant-based monitoring system for the pine-tree lappet moth, *Dendrolimus pini*. As well, the release of a pheromone component of the grass grub *Costelytra zealandica*, phenol, was studied by Harper et al. in 2017 [13].

Field trapping experiments conducted in 2019, 2020, 2021 and 2022, demonstrated that traps baited with precursor A (i.e., **3a**, (*Z*,*Z*)-11,11-dimethoxy-6,9-heneicosadiene) or lure B (i.e., **3b**, (*Z*,*Z*)-6, 9-heneicosadien-11-one ethylene ketal plus acetic acid) performed relatively consistently, capturing significantly more WMTM than traps baited with blank septa in five of seven and six of seven analyses, respectively (Table 2, Table 3, Table 4 and Table 5). In contrast, lure A (precursor A plus acetic acid) and precursor B outperformed the blank septa control in only two of seven analyses. Our results suggest that precursor A is an effective lure for WMTM by itself, breaking down to the WMTM pheromone, (*Z*,*Z*)-6,9-heneicosadien-11-one, without the addition of acetic acid. In contrast, precursor B requires acid to facilitate conversion to the WMTM pheromone. It was noted in the laboratory that the dimethyl acetal in precursor A is cleaved by aqueous acid more quickly than the dioxolane in precursor B, indicating that pheromone **2** could be released more quickly and in higher amounts, at least initially, from precursor A than from precursor B, with or without the addition of acetic acid to the lure.

A possible explanation for precursor A (without acetic acid) catching more moths than lure A (precursor A with acetic acid) in 2021 and 2022 may be because of the logistics of the field studies, (e.g., lures had to be shipped from Fredericton NB to Shubenacadie, NS); once the lures were made (when the pheromone precursor and acetic acid were combined), several days may have passed before the lures were deployed. Consequently, they may have been exhausted and only weakly attractive by the time they were placed in the field. However, the lures were stored at −10 °C wherever possible, and shipped in an ice pack, which would slow down the consumption of pheromone precursor during storage and shipping. The pheromone precursor A without acetic acid, however, could obtain enough acid from traces of carbonic acid in the atmosphere or residues in the rubber septum to release enough pheromone to be an effective lure. Why this effect would occur in 2021and 2022, but not in 2019 and 2020, is unknown to us.

The experiment that tested the effect of lure aging in 2021 showed that lures containing precursor A were as attractive to WMTM after 5 days of aging as they were when freshly deployed (Table 3). Furthermore, traps baited with precursor A lures aged 5 days captured significantly more WMTM than any other lure treatment except fresh precursor A lures, suggesting they should last at least one week in the field. In contrast, the WMTM catch in traps baited with lure B decreased significantly when lures were aged 7 days in the 2020 experiment (Table 3) and by a factor of five when aged 5 days in the 2021 experiment (Table 3). In 2021, the WMTM mean catches in traps baited with lure B were not significantly different from those in traps baited with blank septa, independent of whether lure B was aged 5 days or freshly deployed (Table 3). These results suggest precursor A would last longer in the field than lure B.

In the experiment conducted at Five Islands, NS, in 2020, the mean WMTM catch in traps baited with the commercially available WMTM lure did not differ significantly from any of the other treatments, including blank septa (Table 4). Lure B was the only treatment in this experiment that attracted more WMTM than a blank septum; curiously, traps baited with lure A or precursor A captured no WMTM. However, in 2020, moth populations were lower than expected, possibly due to an outbreak of nuclear polyhedrosis virus (NPV), so results were not optimal (Jeffrey Ogden, personal communication). The composition of the commercial lure was considered proprietary information by Great Lakes IPM, and consequently was not revealed to us. This was consistent with the work of Grant et al. [2], who noted a 69-fold difference between operational traps (commercially available ones) and ketal(acetal)- based traps when they incorporated precursor B into the releasing device that they developed. Unfortunately, we were not able to conclusively prove that either of our formulations consistently outperformed the commercially available lure. For simplicity’s sake, the commercially available lure was only tested at one site in 2020.

The theory behind the vial-based lure was that acetic acid would diffuse into the vial with the pheromone precursor, then be converted to the pheromone, which would then diffuse out of the vial through holes cut in it. The advantage of this device is that the acid does not come into contact with the precursor until the lure is deployed, rather than having the acid and precursor mixed in the septa when the lure is assembled in the lab, then transported to the field. This releasing device did capture moths, but not in significantly different numbers from the controls (see Table 5). Further research and development will be necessary to gauge the scope and usefulness of this releasing device.

Although acid was used to convert the acetal propheromones A and B to the pheromone **2** in our study, it was found by Liu et al. [14] that ultraviolet or simulated sunlight could convert carbonyl compounds protected as *ortho*-nitrophenyl acetals back into the carbonyl compounds, and this system was used in a lure for the diamondback moth *Plutella xylostella* (L.) [14]. A disadvantage to relying on sunlight to release the pheromone is that if the lure is enclosed inside the trap, or out of direct sunlight, conversion to the ketone may not occur optimally. Additionally, ultraviolet light could possibly cause the sensitive skipped diene group of compound **2** undergo rearrangement and decomposition.

Currently, management of WMTM populations involves the applications of insecticides recommended for other defoliators, or commercially available microbial insecticides [15]. As WMTM larvae feed on at least 140 species of woody host plants, including virtually all woody tree and understory species in eastern forests [15], a practical monitoring method for this insect would aid in management programs. Our results indicate that 2 mg of **3a** [(*Z*,*Z*)-11,11-dimethoxy-6,9-heneicosadiene, precursor A] by itself, or 2 mg of **3b** [(*Z*,*Z*)-6, 9-heneicosadien-11-one ethylene ketal, precursor B] plus acetic acid in a red rubber septum are effective lures for WMTM detection. Both treatments performed consistently well as trap lures, catching significantly more WMTM than traps baited with blank septa in five of seven comparisons. However, precursor A may be preferred to lure B, because it does not require the addition of acetic acid and has an operational life of at least 5 days.

This study makes use of the long-range female-produced sex pheromone of WMTM, but full copulatory behavior is induced in WMTM males by pheromones released in the body scales of the female [16,17]. Future work could elaborate upon the effect of semiochemicals in female body scales on long-range male attraction.

## Figures and Tables

**Figure 1 insects-14-00880-f001:**
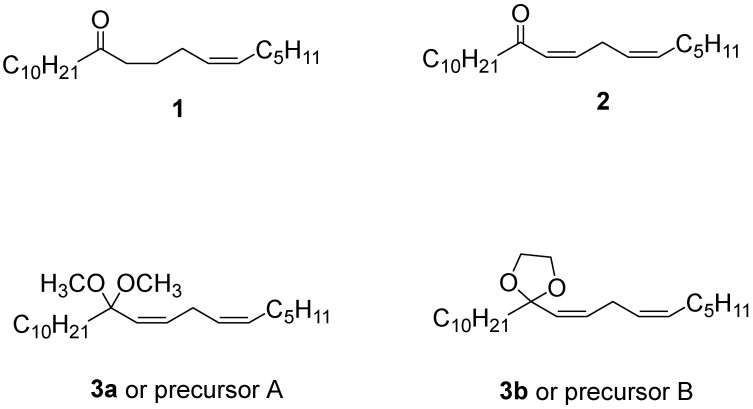
(*Z*)-6-heneicosen-11-one (**1**), (*Z*,*Z*)- 6,9-heneicosadien-11-one (**2**), (*Z*,*Z*)-11, 11-dimethoxy-6, 9-heneicosadiene (**3a**) (precursor A) and (*Z*,*Z*)-6,9-heneicosadien-11-one ethylene ketal (**3b**) (precursor B).

**Figure 2 insects-14-00880-f002:**
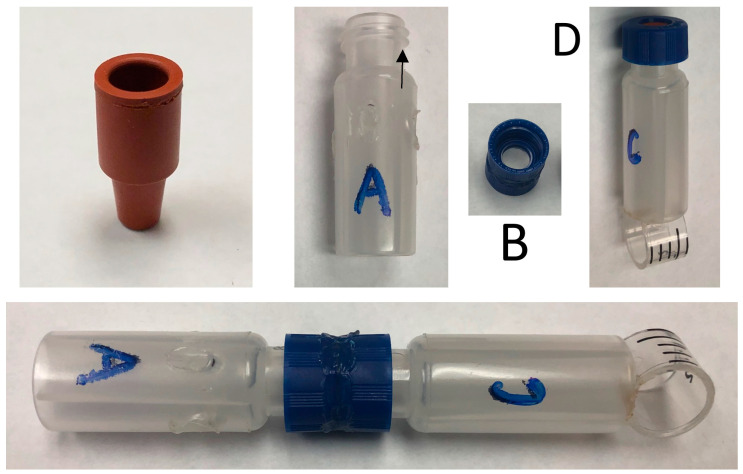
The 5 × 11 mm rubber septum-based releasing device (**above left**) and vial-based releasing device. Vial A contains 2 mg of precursor A and has vent holes drilled into it, component B is made of two welded/fused vial caps with a 0.16 mm low density polyethylene membrane sandwiched between them, and vial C contains 50 µL of acetic acid and is packaged with a septa cap (D) to be removed during assembly (**bottom**).

**Table 1 insects-14-00880-t001:** List of the different experiments testing the effects of various lure treatments and lure age on trap catches of whitemarked tussock moth in Nova Scotia, 2019–2022.

Expt. Number	Lure Treatments	Release Device	Site(s)	Year(s)
1	(*Z*,*Z*)-11, 11-dimethoxy-6, 9-heneicosadiene [=precursor A]	Rubber septum	Five Islands Park	2019
	(*Z*,*Z*)-6,9-heneicosadien-11-one ethylene ketal [=precursor B]	Rubber septum	Earltown	2019, 2020
	Precursor A + acetic acid [=lure A]	Rubber septum	Stillwater	2019
	Precursor B + acetic acid [=lure B]	Rubber septum	Biorachan Road	2021
	Acetic acid	Rubber septum	Riversdale	2022
	Blank	Rubber septum	595 Old Greenfield Rd	2022
			1175 Old Greenfield Rd	2022

2A	Lure A	Rubber septum	Stillwater	2020
	Lure B	Rubber septum		
	Precursor A	Rubber septum		
	Precursor B	Rubber septum		
	Acetic acid	Rubber septum		
	Blank septa	Rubber septum		
	Lure A aged 1 week at room temperature	Rubber septum		
	Lure B aged 1 week at room temperature	Rubber septum		
2B	Lure A	Rubber septum	Dalhousie Mtn.	2021
	Lure B	Rubber septum		
	Precursor A	Rubber septum		
	Precursor B	Rubber septum		
	Acetic acid	Rubber septum		
	Blank	Rubber septum		
	Lure A aged 5 days at room temperature	Rubber septum		
	Lure B aged 5 days at room temperature	Rubber septum		
	Precursor A aged 5 days at room temperature	Rubber septum		
	Precursor B aged 5 days at room temperature	Rubber septum		

3	Lure A	Rubber septum	Five Islands Park	2020
	Lure B	Rubber septum		
	Precursor A	Rubber septum		
	Precursor B	Rubber septum		
	Acetic acid	Rubber septum		
	Blank	Rubber septum		
	WMTM lure (Great lakes IPM)	Rubber septum		

4	Lure A	Rubber septum	Proudfoot Rd.	2021
	Lure B	Rubber septum		
	Precursor A	Rubber septum		
	Precursor B	Rubber septum		
	Acetic acid	Rubber septum		
	Blank	Rubber septum		
	50 μL acetic acid [AA]	Vial-based lure		
	2 mg pheromone precursor A [PA]	Vial-based lure		
	2 mg pheromone precursor A + 50 μL acetic acid [PA + AA]	Vial-based lure		

**Table 2 insects-14-00880-t002:** Mean catch of WMTM (±SE) in Multipher II traps (in 2019, 2020 and 2022) or Multipher I and II traps (2021) baited with six different lure treatments (Experiment 1) conducted in Nova Scotia in 2019 (three sites), 2020 (one site), 2021 (one site), and 2022 (three sites). Data pooled from 2019 plus 2020 were analyzed separately from those in 2021 and 2022 due to significant site–year*treatment interactions. Within years, means followed by different letters were significantly different (Tukey–Kramer test, *p* ≤ 0.05).

Year	2019 + 2020		2021		2022	
Treatment	Mean	SE	Mean	SE	Mean	SE
Lure A	14.9	3.85 ^a^	1.33	0.76 ^bc^	1.67	0.56 ^b^
Lure B	3.33	0.87 ^b^	5.33	2.30 ^ab^	1.21	0.43 ^b^
Precursor A	2.00	0.63 ^b^	12.0	1.46 ^a^	9.96	2.18 ^a^
Precursor B	0.25	0.14 ^c^	5.83	4.45 ^b^	0.08	0.06 ^c^
Acetic acid	0.42	0.94 ^c^	0	0 ^c^	0	0 ^c^
Blank septa	0	0 ^c^	0	0 ^c^	0	0 ^c^

**Table 3 insects-14-00880-t003:** Mean catch of WMTM (±SE) in Multipher I or II traps baited with different lure treatments testing the effect of aging specific lures for 7 days (Experiment 2A, Stillwater, Nova Scotia, 2020) or 5 days (Experiment 2B, Dalhousie Mountain, Nova Scotia, 2021). Means followed by different letters were significantly different (Friedman’s test, ANOVA on data converted to ranks, followed by Ryan–Einot–Gabriel–Welsh multiple range test, *p* ≤ 0.05).

	Expt 2A		Expt 2B	
Treatment	Mean	SE	Mean	SE
Lure A	0.13	0.13 ^b^	2.00	0.85 ^c^
Lure B	1.38	0.53 ^a^	5.00	2.89 ^bc^
Precursor A	0.25	0.16 ^b^	8.00	1.25 ^ab^
Precursor B	0.13	0.13 ^b^	1.14	0.63 ^c^
Acetic acid	0	0 ^b^	0	0 ^c^
Blank septa	0	0 ^b^	0	0 ^c^
Lure A aged	0	0 ^b^	2.43	2.27 ^c^
Lure B aged	0	0 ^b^	0.71	0.36 ^c^
Precursor A aged	-	-	13.6	2.85 ^a^
Precursor B aged	-	-	1.71	0.84 ^c^

**Table 4 insects-14-00880-t004:** Mean catch of WMTM (±SE) in Multipher II traps baited with different lure treatments in experiment 3 conducted in Five Islands Park, Nova Scotia in 2020. Means followed by different letters were significantly different (Tukey–Kramer test, *p* ≤ 0.05).

Treatment	Mean	SE
Lure A	0	0 ^b^
Lure B	3.25	1.65 ^a^
Precursor A	0	0 ^b^
Precursor B	0.25	0.14 ^b^
Acetic acid	0	0 ^b^
Blank septa	0	0 ^b^
WMTM lure	0.50	0.29 ^ab^

**Table 5 insects-14-00880-t005:** Mean catch of WMTM (±SE) in Multipher II traps baited with nine different lure treatments in experiment 4 conducted in Proudfoot Road, Nova Scotia in 2021, testing vial-based lures. Means followed by different letters were significantly different (Tukey–Kramer test, *p* ≤ 0.05).

Treatment	Mean	SE
Lure A	0.78	0.43 ^c^
Lure B	12.5	6.17 ^ab^
Precursor A	24.4	6.55 ^a^
Precursor B	8.25	3.67 ^ab^
Acetic acid	0	0 ^c^
Blank septa	0	0 ^c^
Vial- acetic acid (AA)	0	0 ^c^
Vial-precursor A (PA)	0.63	0.50 ^c^
Vial-PA + AA	1.29	0.61 ^bc^

## Data Availability

The data presented in this study are available on request from the corresponding author.

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
