# Peer review of "Field Test of the Propheromones of the Whitemarked Tussock Moth (WMTM) Orgyia leucostigma (J.E. Smith) (Lepidoptera: Erebidae)"

_insects, 2023, doi:10.3390/insects14110880_

Round 1

Reviewer 1 Report

Comments and Suggestions for Authors

              The reviewer has read with interest the manuscript entitled as ‘Field test of the propheromones of the whitemarked tussock moth (WMTM) Orgyia leucostigma (J.E. Smith) (Lepidoptera: Erebidae)’ submitted by Peter Mayo, Sumudu Deepa Abeysekera, Peter J. Silk, David I. MaGee, Gaetan Leclair, Jon Sweeney and Jeffrey Ogden to INSECTS. The authors attempted to improve a known lure based on the pheromone of this insect with the aim to use this lure as a widespread monitoring / detection method by field tests. However, some serious problems are present in this manuscript.

1.         The experiments were conducted in nine areas and the data obtained in these different areas were statistically processed as a whole. Did each area have area-specific characteristics related to WMTM? If so, the data should be processed separately and the conclusion might change by some chance.

2.         It is somewhat difficult to understand the devices which the authors used without their figures. It is better to show diagrammatically the experimental devices.  

3.         As the Names of some devices such as Multipher I and II which the authors used in the experiments were trade ones, some reader may not understand the devices. It is better to explain shortly their shapes

4.         The reviewer did not find Figure 2, which was indicated at Line 97.

5.         Though the results were shown by four tables, significant differences between results of different treatments seem inconsistent between some results of different years. How did the authors estimate differences of the effectivities of the treatments and devices from inconsistent results?

6.         The results were shown in tables in this manuscript, the reviewer recommends to show these with figures(graphs) for ease of comprehension.

7.         The authors commented impurities and degenerations of precursors as the reasons of difference of the numbers of caught moths (Lines 232-245). When the authors concerned about such situations, the authors should not use the data related to such situations.

8.         Summary and Abstract should be written in order for readers to understand broadly the contents of the paper. Those of this manuscript only partially covered contents of the manuscript.

9.         The reviewer could not understand two formulae y = lure + site-year + block (Line145) and y = lure + block (Line147).  

Author Response

Thanks for a thorough review. I have attempted to address the comments (see below for explanations / corrections):

  1. The experiments were conducted in nine areas and the data obtained in these different areas were statistically processed as a whole. Did each area have area-specific characteristics related to WMTM? If so, the data should be processed separately and the conclusion might change by some chance.

We replicated experiment 1 in more than one site and year to provide a more robust test of our treatments and pooled data from different site-years to provide more statistical power and less chance of committing a type II error.  However, we agree with the reviewer’s point that WMTM response to lure treatments may have differed among site-years. For this reason, we first tested for interactions between lure and site-year using the model: y = lure + site-year + lure*site-year + block(site-year), with both lure and site-year considered as fixed effects and blocks as random effects. When the site-year interaction was not significant, data were pooled and analyzed using the following model: y = lure + site-year + block(site-year), with lure treatments fixed and both site-years and blocks random.  We had explained this in a previous version of the manuscript but had removed it from our journal submission in an effort to reduce the amount of text in the Methods. However, in the interests of clarity we have reinserted the following text on lines 162 to 167 in the revised manuscript:

“For experiment 1, we first ran a preliminary analysis to test for significant interactions between lure treatment and site-year, using the model: y = lure + site-year + lure*site-year + block(site-year), with both lure and site-year considered as fixed effects and blocks as random effects. When the site-year interaction was not significant, data were pooled and analyzed using the following model: y = lure + site-year + block(site-year), with lure treatments fixed and both site-years and blocks random.” 

  1. It is somewhat difficult to understand the devices which the authors used without their figures. It is better to show diagrammatically the experimental devices.  

A new figure (2) was added which illustrates both the septum-based and vial-based releasing devices.

  1. As the Names of some devices such as Multipher I and II which the authors used in the experiments were trade ones, some reader may not understand the devices. It is better to explain shortly their shapes.

Good point. We now provide a link to the manufacturer’s webpage that has images of both trap types (https://solida.quebec/produit/multipher-i-ii-and-iii-product-no-40bc001/?lang=en) on lines 126-127 of the revised manuscript.

  1. The reviewer did not find Figure 2, which was indicated at Line 97.

See response to point 2 above.

  1. Though the results were shown by four tables, significant differences between results of different treatments seem inconsistent between some results of different years. How did the authors estimate differences of the effectivities of the treatments and devices from inconsistent results?

Yes, there were differences among site-years in the relative performance of some of our lure treatments and potential reasons for these differences are suggested in the Discussion. However, we were able to discern general trends in our results, as we state in the first lines of the Discussion:

“Field trapping experiments conducted in 2019, 2020, 2021 and 2022, demonstrated that traps baited with precursor A (i.e., 3a, (Z, Z)-11, 11-dimethoxy-6, 9-heneicosadiene) or lure B (i.e., 3b, 2-((Z, Z)-1, 4-decadienyl)-2-ndecyl-1, 3-dioxolane plus acetic acid) performed relatively consistently, capturing significantly more WMTM than traps baited with blank septa in 5 of 7 and 6 of 7 analyses, respectively (Tables 2-5). In contrast, lure A (precursor A plus acetic acid) and precursor B each outperformed the blank septa control in only 2 of 7 analyses.  Our results suggest that precursor A is an effective lure for WMTM by itself, breaking down to the WMTM pheromone, (Z, Z)-6, 9-heneicosadien-11-one, without the addition of acetic acid. In contrast, compound precursor B requires acid to facilitate conversion to the WMTM pheromone.”

  1. The results were shown in tables in this manuscript, the reviewer recommends to show these with figures(graphs) for ease of comprehension.

We respectfully disagree. The Tables show our results clearly.

  1. The authors commented impurities and degenerations of precursors as the reasons of difference of the numbers of caught moths (Lines 232-245). When the authors concerned about such situations, the authors should not use the data related to such situations.

It seems safe to omit from our manuscript the comment which speculates on differences in impurities in different batches of pheromone precursor from year to year, as the same methods were used every year.

  1. Summary and Abstract should be written in order for readers to understand broadly the contents of the paper. Those of this manuscript only partially covered contents of the manuscript.

Respectfully, myself and the other authors feel that the simple summary and abtract were descriptive and concise enough.

  1. The reviewer could not understand two formulae y = lure + site-year + block (Line145) and y = lure + block (Line147).  

This is standard terminology for statistical analyses using analysis of variance and generalized linear models. In experiment 1, replicated in more than one site year, the model was y = lure + site-year + block(site-year), where “y” is the response variable, i.e., trap catch of WMTM in this case, “lure” is our lure treatment, “site-year” is the site and year where the experiment was replicated, and block(site-year) is a spatial replicate of treatments nested within site-years. The simpler model (y = lure + block) was used for experiments that were run in only one site and year.  Each model also implicitly includes random error (any variation in catch not accounted for by lure, site-year or block) but this is not usually explicitly stated.  Lure is a “fixed effect”, i.e., specific lure types that we tested for their effect on trap catch of WMTM, whereas site-year and block are random factors, included in the model to account for any variation in WMTM trap catch they may explain, thus reducing the unexplained random error in the model. 

Reviewer 2 Report

Comments and Suggestions for Authors

My comments are shown on the attached PDF file

1-The version that I had access to did not show Figure 2

2-The labeling of cuemicals (Figure 1 , 2, 3a, 3b) needs to be revised?

3-Treatment names (Lure A, Precursor A, etc) should be revised

4-Statistics: If the model does not converge should try models that deal with zero-inflation and overdispersion.

5-Please, put Supplementary Table 1 in the main text and complete that table so that septum vs vial is clearly stated (and revise treantment names). The experiments are so diverse and variable that an effor should be made to ease the task to the readers

6-The discussion is too long, perhaps a bit speculative. Could be trimmed

Author Response

Thanks for a thorough and generally positive review. First to address the written comments and suggestions:

1-The version that I had access to did not show Figure 2

I now include Figure 2 which was left out in oversight.

2-The labeling of chemicals (Figure 1 , 2, 3a, 3b) needs to be revised?

I attempted to disambiguate the naming of compounds. After Figure 1, I changed 3a to precursor A every time it appeared in the text, likewise 3b to precursor B. Although 3b is called out before 3a in the text, (so, as the reviewer states, shouldn’t it be called 3a?) It seemed less ambiguous to us to call 3a precursor A and 3b precursor B because in all of our experimental lab and field notes this is the convention we used, ie. A corresponds to a and B to b.

3-Treatment names (Lure A, Precursor A, etc) should be revised.

See point 2 above. It is stated in the new Table 1 (formerly supplementary Table 1) text that the treatment with both precursor A and acid in the septum is called “Lure A” whereas precursor A in the septum with no acid is simply called “Precursor A”, likewise for Lure B and precursor B.

4-Statistics: If the model does not converge should try models that deal with zero-inflation and overdispersion.

We chose the non-parametric Friedman test due to the significant number of zeros in our data that violate parametric assumptions, even with zero inflation. The Friedman test's non-parametric nature allows us to avoid making assumptions about data distribution, ensuring the robustness of our analysis.

5-Please, put Supplementary Table 1 in the main text and complete that table so that septum vs vial is clearly stated (and revise treatment names). The experiments are so diverse and variable that an effort should be made to ease the task to the readers.

Supplementary Table 1 has been moved to the main text. A line has been added in the table heading stating that all data is for rubber septum-based lures, unless polypropylene vials are indicated.

6-The discussion is too long, perhaps a bit speculative. Could be trimmed

An attempt has been made to remove unessential material from the discussion.

Response to Reviewer 2’s comments in margins of the manuscript

Page 1: Sumudu is the author’s first name, but she goes by Deepa.

Page 2:

-(Z, Z)-6, 9-heneicosadien-11-one is actually part of the whitemarked tussock moth female-produced pheromone blend. I have added a sentence in the introduction to state this on line 55 of the revised manuscript.

-see points 2 and 3 above regarding my attempt to disambiguate the naming of compounds.

Page 3:

-I have given septum and vial-based releasing devices separate headings in the Materials and Methods section.

-NS has been replaced with Nova Scotia, Canada.

-Figure 2 has been added, which illustrates vials A and C and the connecting device B.

-The thickness of the polyethylene membrane is 0.16 mm, not 6 mm as I mistakenly specified (it is actually 6 mil or 6 thousandths of an inch). This is thin enough to be permeable. The membrane is in fact made of polyethylene, not polypropylene like the vials.

-I changed 4 x 1/8” to “four 1/8 inch wide breather holes”.

-As stated previously, Supplementary Table 1 has been moved to the main text. A line has been added in the table heading stating that all data is for rubber septum-based lures, unless polypropylene vials are indicated.

Overdispersion and zero inflation: -We chose the non-parametric Friedman test due to the significant number of zeros in our data that violate parametric assumptions, even with zero inflation. The Friedman test's non-parametric nature allows us to avoid making assumptions about data distribution, ensuring the robustness of our analysis.

Page 6:

Did you test for interactions? Yes, we tested for interactions between lure treatment and site-year and only pooled data from site-years for which there was no significant interaction. We now explain this on lines 162-167 in the revised manuscript. Please also see our response to comment 1 from Reviewer 1.

Page 7:

-the sentence beginning “Further support for this is our finding…” has been changed to “It was noted anecdotally in the laboratory that the dimethyl acetal is cleaved by aqueous acid more quickly than the dioxolane in precursor B, indicating that pheromone 2 could be released more quickly and in higher amounts, at least initially, from precursor A than from precursor B, with the addition of acetic acid to the lure. (lines 253-256 of revised manuscript).

Page 9:

-the sentence beginning “Trapping data, in spreadsheet form, are not publically available…” has been omitted.

Round 2

Reviewer 1 Report

Comments and Suggestions for Authors

              The reviewer has read the revised manuscript entitled as ‘Field test of the propheromones of the whitemarked tussock moth (WMTM) Orgyia leucostigma (J.E. Smith) (Lepidoptera: Erebidae)’ submitted to INSECTS. The authors attempted to improve known lures based on the pheromones of this insect with the aim to use this lure as a widespread monitoring / detection method by field tests. The reviewer thinks that the authors revised almost properly the manuscript. Though the editor pointed out the number of references is too few, the reviewer thinks no problems.

Author Response

Thanks for the positive review. As no changes were suggested, please see the response to the second reviewer.

Reviewer 2 Report

Comments and Suggestions for Authors

Author names: What is the point of puting a name in parenthesis? Every time the paper is cited the parenthesis must be included. Why make things more difficult? It is already complicated when a person is called: "A.B. Cccc-Dddd", whay then make it more complicated with (A) B. Cccc-Dddd?

Table 1, methodologogy. Please include a column indicating the type of substrate used in each case (septum or vial). Although only the last 2 or 3 lines are vials and the rest are septa, I think it will help readers have this information in the table.

The optical quality of the septum and vial pictures is low. Can you provide better pictures?

Results: Raw data are necessary so that others can reanalyze them. Please place them in a repository (or as an attachment) and provide a link.

Author Response

Thanks for the thorough review.

To answer the reviewers comments:

Author names: What is the point of puting a name in parenthesis? Every time the paper is cited the parenthesis must be included. Why make things more difficult? It is already complicated when a person is called: "A.B. Cccc-Dddd", whay then make it more complicated with (A) B. Cccc-Dddd?

-the parentheses were removed from Sumudu Deepa Abeysekera's name. In all previous publications, she is known as Sumudu Deepa Abeysekera, so this is the name used in this paper.

Table 1, methodologogy. Please include a column indicating the type of substrate used in each case (septum or vial). Although only the last 2 or 3 lines are vials and the rest are septa, I think it will help readers have this information in the table.

-this table was reformatted, with a column indicating the substrate used (septum or vial).

The optical quality of the septum and vial pictures is low. Can you provide better pictures?

-Figure 2 was improved, with an improved photograph of the septum-based releasing device, and more in-depth, higher quality photos illustrating the vial-based releasing device.

Results: Raw data are necessary so that others can reanalyze them. Please place them in a repository (or as an attachment) and provide a link.

-the 4 datasets from the 4 years of the study (2019-2022) were added as zipped supplementary files.